# Factors associated with international humanitarian aid appeal for disasters from 1995 to 2015: A retrospective database study

Lenard Cheng[1]*, Attila J. Hertelendy[1,2], Alexander Hart[1,3,4], Lawrence Siu-Chun Law[5], Ryan Hata[1], Georgina Nouaime[1], Fadi Issa[1], Lina Echeverri[6], Amalia Voskanyan[1], Gregory R. Ciottone[1]

1 Beth Israel Deaconess Medical Center Fellowship in Disaster Medicine, Harvard Medical School, Boston, Massachusetts, United States of America, 2 Department of Information Systems and Business Analytics, College of Business, Florida International University, Miami, Florida, United States of America, 3 Department of Emergency Medicine, Hartford Hospital, Hartford, Connecticut, United States of America, 4 University of Connecticut School of Medicine, Farmington, Connecticut, United States of America, 5 National University Hospital, Singapore, Singapore, 6 Center for Research and Training in Disaster Medicine, Humanitarian Aid and Global Health, Universitá del Piemonte Orientale, II Piano, Novara, Italy

* lcheng2@bidmc.harvard.edu

**Data Availability Statement:** All relevant data are within the paper and its Supporting information files.

## Abstract

### Introduction

International humanitarian aid during disasters should be needs-based and coordinated in response to appeals from affected governments. We identify disaster and population factors associated with international aid appeal during disasters and hence guide preparation by international humanitarian aid providers.

### Methods

In this retrospective database analysis, we searched the Emergency Events Database for all disasters from 1995 to 2015. Disasters with and without international aid appeals were compared by location, duration, type of disaster, deaths, number of people affected, and total estimated damage. Logistic regression was used to examine the association of each factor with international aid appeal.

### Results

Of 13,961 disasters recorded from 1995 to 2015, 168 (1.2%) involved international aid appeals. Aid appeals were more likely to be triggered by disasters which killed more people (OR 1.29 [95% confidence interval (CI) 1.02–1.64] $\log_{10}$ persons), affected more people (OR 1.85 [95%CI 1.57–2.18] / $\log_{10}$ persons), and occurred in Africa (OR 1.67 [95%CI 1.06–2.62]). Earthquakes (OR 4.07 [95%CI 2.16–7.67]), volcanic activity (OR 6.23 [95%CI 2.50–15.53]), and insect infestations (OR 12.14 [95%CI 3.05–48.35]) were more likely to trigger international aid appeals. International aid appeals were less likely to be triggered by disasters which occurred in Asia (OR 0.46 [95%CI 0.29–0.73]) and which were transport accidents (OR 0.12 [95%CI 0.02–0.89]).

**Funding:** The authors received no specific funding for this work.

**Competing interests:** The authors have declared that no competing interests exist.

## Conclusion

International aid appeal during disasters was associated with greater magnitude of damage, disasters in Africa, and specific types of disasters such as earthquakes, volcanic activity, and insect infestations. Humanitarian aid providers can focus preparation on these identified factors.

## Introduction

Disasters claim an average of 61,000 lives and disrupt 201 million others, costing US $151 billion every year [1]. The global impact of these disasters has impelled the United Nations (UN) Deputy Secretary-General Amina Mohammed to declare disasters a "spiral of self-destruction" for humanity [2], and more than 250 journals to simultaneously signal a climate crisis of "supreme concern" needing "all hands on deck" [3]. Low-income countries are disproportionately affected by disasters: not only are death rates comparatively higher, but also socioeconomic resilience is hampered by the cycle of poverty, vulnerability, and loss [4, 5]. A resultant 274 million people are estimated by the UN to require humanitarian assistance and protection in 2022 [6].

International humanitarian aid is a major source of assistance. In 2019 alone, the UN committed US $18 billion in contributions to more than 117 million people requiring humanitarian aid, and the International Federation of Red Cross Red Crescent Societies deployed more than 400 professionals to disaster-stricken communities as part of a humanitarian system that "has never been more vital" [7, 8].

However, international aid can have unintended consequences [9], as exemplified by the 2010 Haitian earthquake. Disorganized, ill-prepared, and uninvited international humanitarian aid can create additional burdens to the affected population, and responders can even compete for limited local resources for food and shelter [10, 11].

In response, organizations such as the UN Office for the Coordination of Humanitarian Affairs (OCHA) and World Health Organization (WHO) Emergency Medical Teams (EMT) Initiative published guidelines for international humanitarian providers [9, 12]. These guidelines aim to prepare for a "needs-driven response", and reserve activation until after there have been "initial impact assessments by relevant local authorities" and a "formal request for international assistance" [9, 13].

However, to our knowledge, no study has identified disasters likely to trigger formal requests for international aid. Consequently, international humanitarian providers are limited in tailoring preparation for a needs-driven response.

This study compares worldwide disasters with and without formal appeals for international aid, and identifies disaster and population characteristics associated with international aid appeals. Preparation by international humanitarian aid providers for needs-driven response will be better guided by these findings.

## Materials and methods

### Study design and definitions

This was a retrospective database search using the Emergency Events Database (EM-DAT) for worldwide disasters from 1995 to 2015 [14]. The EM-DAT is a public free-access database provided by the Centre for Research on the Epidemiology of Disasters, compiled from sources such as UN agencies, governments, the International Federation of Red Cross and Red

Crescent Societies, non-government agencies, insurance companies, and news outlets [15]. Events are included if they meet the following criteria: (1) 10 or more human deaths, (2) 100 or more people affected/injured/homeless, (3) declaration of a state of emergency by the affected country, or (4) international aid appeal by the affected country [14]. Events dating before 1995 were excluded to minimize bias from the paucity of records before and at the start of the database's 1988 launch [16, 17]. Events dating after 2015 were excluded after correspondence with and at the advice of the database's authors, due to incomplete data entry pertaining to international aid appeal at the time of writing.

International aid appeal is defined by the database authors as "any request for international assistance" [15]. Since the database data entry for this variable is binary, blank fields were interpreted as absence of international aid appeal. Types of disasters follow the classification from and are defined by the Integrated Research on Disaster Risk Peril Classification and Hazard Glossary [18]. Specifically, the "main event" classification level (earthquake, volcanic activity, flood) was chosen to aggregate similar hazards and preparation principles, and matched terminology used in most disaster reports. All other classifications and definitions are defined by the EM-DAT as per their database guidelines and glossary [16, 19].

## Data collection and statistical analysis

Disasters with and without international aid appeals were described and compared using the following factors of location by continent, duration in days, type of disaster, total deaths, total number of people affected, and total estimated damage in US $ million (inflation-adjusted to the year of 2021). Logarithmic transformation to near normality was performed before analyses for disaster duration, total deaths, total number of people affected, and total estimated damage. Mann-Whitney U Test was used to examine the overall trend of international aid appeals to total number of disasters ratio. $X^2$ test was used for categorical variables. For variables with multiple categories (year, location and types of disasters), the p-value cut-offs from the $X^2$ test were Šidák corrected to 0.01 and 0.002 respectively. T-test was used for continuous variables. Factors with $p < 0.05$ (or Šidák corrected p) were included in the logistic regression to examine their independent association with international aid appeal. Odds ratios (OR) and 95% confidence intervals (95%CI) were reported. Otherwise, the cut-off for significant p-values was set at 0.05. Statistical analyses were carried out using IBM SPSS Statistics 26.0 (Armonk, NY: IBM Corp).

Missing data analyses showed that the missing data was non-random. Disasters with missing data in duration and total deaths were associated with more international aid appeals ($p < 10^{-11}$ and 0.017 respectively), whereas disasters with missing data in total number of people affected and total estimated damage were associated with fewer international aid appeals ($p < 10^{-10}$ and $p < 10^{-20}$ respectively). Excluding all records with missing data from the regression caused systematic bias. Manual imputation for these records with missing data was neither feasible nor reliable. Substituting all missing data with mean or median incorrectly reduced the variance of the variables. Thus, multiple imputation was used to impute missing data for all analyses (fully conditional specification, 10 iterations, predictive mean matching, 10 imputations, SPSS 26.0).

Ethics approval was not required for this analysis as the primary dataset is publicly available. Potentially sensitive data involving deaths and injury are not individual patient-level, and are compiled from published sources. Patients or the public were not involved in the design, or conduct, or reporting, or dissemination plans of our research.

## Results

A total of 13,961 disasters had been recorded from 1995 till 2015, of which 168 (1.2%) involved international aid appeals (Fig 1). In 2004, a spike of 20 (2.6%, $p < 0.001$)) disasters with

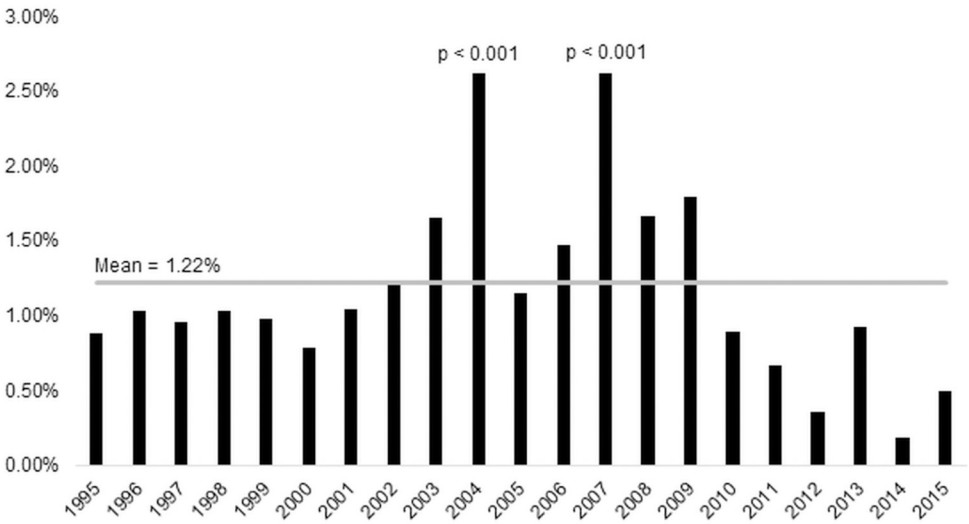

**Fig 1. Percentage of total disasters triggering international aid appeal by years.** The grey horizontal line indicates the mean percentage of 1.22%. Years 2004 and 2007 register significantly more international aid appeals. Šidák corrected p cutoff = 0.002 for years as each was compared to other years by $X^2$ test. Overall trend was not significantly increasing or decreasing (Mann-Whitney U Test: p = 0.496).

international aid appeals was associated with 6 insect infestation-type disasters. This number was anomalous when compared with all other years which documented less than one insect infestation per year. In 2007, 11 floods made up a majority of the 19 (2.6%, p < 0.001) disasters with international aid appeals. The proportion of total disasters with international aid appeals was otherwise stable from 1995 till 2015 (p = 0.496).

Geometric means for duration was 2.5 (95% confidence interval [CI] 2.4–2.6) days, total number of people deaths was 16.9 (95%CI 16.4–17.3) people, total number of people affected was 592.4 (95%CI 554.4–632.9) people, and estimated damage was US $29.0 (95% CI 27.6–30.4) million. 8,390 (60.1%) were natural disasters by database classification. Most disasters occurred in Asia (5,727 [41.0%]), followed by Africa (3,500 [25.1%]), the Americas (2,716 [19.5%]), and Europe (1,663 [11.9%]), whereas Oceania had the least (355 [2.5%]).

Table 1 describes the comparison of disasters with and without international aid appeal in terms of the duration of disaster, total deaths, total number of affected persons, and total estimated damage, continent, and type of disaster. Disasters triggering international aid appeals were longer in duration (9.36 vs 2.47 days, p < $10^{-19}$), killed more people (37.63 vs 16.71 persons, p < $10^{-5}$), affected more people (75,125.98 vs 558.43 persons, p < $10^{-93}$), and caused more damage (US $60.95 vs 28.71 million, p < 0.003).

Table 2 describes the logistic regression analysis which suggested more deaths (OR 1.29 [95%CI 1.02–1.64] / $\log_{10}$ persons), more people affected (OR 1.85 [95%CI 1.57–2.18] / $\log_{10}$ persons), disasters occurring in Africa (OR 1.67 [95%CI 1.06–2.62]), earthquakes (OR 4.07 [95%CI 2.16–7.67]), volcanic activity (OR 6.23 [95%CI 2.50–15.53]), and insect infestations (OR 12.14 [95%CI 3.05–48.35]) to be positively associated with international aid appeal. Disasters occurring in Asia (OR 0.46 [95%CI 0.29–0.73]) and transport accidents (OR 0.12 [95%CI 0.02–0.89]) were associated with lower likelihood of an international aid appeal. However, duration of disaster and total estimated damage were not independently associated with international aid appeal.

**Table 1. Comparison of disasters with and without international aid appeal by duration of disaster, total deaths, total number of affected persons, and total estimated damage, continent, and type of disaster.**

| Factors | No aid appeal | With aid appeal | p |
|---|---|---|---|
| | **Geometric Mean (95% Confidence Interval)** | | |
| Duration (days) | 2.47 (2.40–2.53) | 9.36 (7.02–12.47) ↑ | $< 10^{-19}$ |
| Total death (persons) | 16.71 (16.21–17.22) | 37.63 (26.51–53.41) ↑ | $< 10^{-5}$ |
| Total number of people affected (persons) | 558.43 (523.56–595.63) | 75,125.98 (47,778.79–118,125.91) ↑ | $< 10^{-93}$ |
| Total estimated damage (US $ million inflation-adjusted to year 2021) | 28.71 (22.79–36.16) | 60.95 (37.27–99.68) ↑ | 0.003 |
| **Continent** | **No aid appeal** | **With aid appeal** | **p** |
| Africa | 3,431 (98.0%) | 69 (2.0%) ↑ | $< 10^{-5}$ |
| Americas | 2,681 (98.7%) | 35 (1.3%) | 0.65 |
| Asia | 5,677 (99.1%) | 50 (0.9%) ↓ | 0.003 |
| Europe | 1,655 (99.5%) | 8 (0.5%) ↓ | 0.004 |
| Oceania | 349 (98.3%) | 6 (1.7%) | 0.394 |
| Total | 13,793 (98.8%) | 168 (1.2%) | |
| **Disaster type** | **No aid appeal** | **With aid appeal** | **p** |
| Animal accident | 1 (100.0%) | 0 (0.0%) | 0.912 |
| Complex disasters | 9 (90.0%) | 1 (10.0%) | 0.011 |
| Drought | 331 (92.7%) | 26 (7.3%) ↑ | $< 10^{-25}$ |
| Earthquake | 556 (97.0%) | 17 (3.0%) ↑ | $< 10^{-4}$ |
| Epidemic | 1,027 (99.3%) | 7 (0.7%) | 0.107 |
| Extreme Temperature | 410 (99.5% | 2 (0.5%) | 0.175 |
| Flood | 3,076 (98.0%) | 64 (2.0%) ↑ | $< 10^{-5}$ |
| Glacial lake outburst | 0 | 0 | NA |
| Impact | 1 (100.0%) | 0 (0.0%) | 0.912 |
| Industrial accident | 935 (99.8%) | 2 (0.2%) | 0.004 |
| Insect infestation | 18 (72.0%) | 7 (28.0%) ↑ | $< 10^{-34}$ |
| Landslide | 387 (98.5%) | 6 (1.5%) | 0.551 |
| Mass movement (dry) | 11 (100.0%) | 0 (0.0%) | 0.714 |
| Miscellaneous | 847 (99.9%) | 1 (0.1%) | 0.003 |
| Storm | 2,050 (98.8%) | 24 (1.2%) | 0.834 |
| Transport accident | 3,775 (100.0%) | 1 (0.0%) ↓ | $< 10^{-14}$ |
| Volcanic activity | 105 (94.6%) | 6 (5.4%) ↑ | $< 10^{-4}$ |
| Wildfire | 254 (98.4%) | 4 (1.6%) | 0.606 |
| Total | 17,275 (99.0%) | 183 (1.0%) | |

Šidák corrected p cutoff = 0.01 for continents as each was compared to all other continents by $X^2$ test; Šidák corrected p cutoff = 0.002 for disaster type as each was compared to all other disasters types by $X^2$ test. Significant p-values were underlined and the corresponding variables were entered in the logistic regression.

## Discussion

The core mission of humanitarian action is to preserve life and protect communities [6]. However, history has repeatedly demonstrated failure to put this mission into practice, when unprepared and uninvited actors inflict additional burden on local resources and chaos that undermines local systems [10]. For this reason, it is imperative for international humanitarian aid to support local surge capacity rather than invade the sovereignty of the afflicted community, remaining respectful of local beliefs and values [9]. Our findings contribute by supporting local government request as prerequisite to international humanitarian response and providing a guide for specific preparation.

**Table 2. Logistic regression of factors associated with international aid appeal.**

| Factors | Odds ratio | 95% Confidence Interval | p |
|---|---|---|---|
| Duration ($\log_{10}$ days) | 1.31 | 0.98–1.76 | 0.07 |
| Total death ($\log_{10}$ persons) | 1.29 | 1.02–1.64 | 0.034 |
| Total number of people affected ($\log_{10}$ persons) | 1.85 | 1.57–2.18 | $< 10^{-12}$ |
| Total estimated damage (log10 US $ million adjusted) | 0.82 | 0.49–1.39 | 0.471 |
| Africa | 1.67 | 1.06–2.62 | 0.026 |
| Asia | 0.46 | 0.29–0.73 | 0.001 |
| Europe | 0.58 | 0.27–1.25 | 0.166 |
| Drought | 1.05 | 0.52–2.10 | 0.895 |
| Earthquake | 4.07 | 2.16–7.67 | $< 10^{-4}$ |
| Flood | 1.37 | 0.90–2.09 | 0.14 |
| Transport accidents | 0.12 | 0.02–0.89 | 0.038 |
| Volcanic activities | 6.23 | 2.50–15.53 | $< 10^{-4}$ |
| Insect infestation | 12.14 | 3.05–48.35 | $< 0.001$ |

Intuitively, disasters that killed and affected more people were more likely to trigger international aid appeals. Of these disasters with appeals, earthquakes accounted for 6 of the top 10 by fatality (the deadliest of which was the 2003 Bam earthquake) and will be discussed later in this section. On the other hand, floods accounted for 2 out of the leading 4 that affected tens of millions of people. The 2004 Bangladesh monsoon floods affected 36 million people, destroyed 2 million acres of agricultural land, and cost US $2.3 billion [20]. International humanitarian aid, headlined by a UN appeal to international donors for US $210 million, brought food supplies (the UN World Food Programme distributed rations to almost 2 million people within the start of relief operations [21]) and medical teams. Close to 30 teams were deployed by the International Federation of the Red Cross and Red Crescent Societies (IFRC) alone [22]. Agriculture-dependent populations like Bangladesh are especially vulnerable to the devastation that floodwaters wreak on farmlands and the long-term strain on livelihood [23]. International humanitarian efforts must hence be sustainable by transitioning from response to recovery. For example, the UN World Food Programme Food-for-Work demonstrates a method in which locals receive rice in payment for labor rebuilding local infrastructure [24, 25]. Ultimately, the international community must not "forget" long-drawn humanitarian crises [26], as has befallen war-torn Afghanistan in the wake of shifting media and political attention to the war in Ukraine [27].

The finding that disasters originating in Africa are associated with international aid appeals is corroborated by a 2022 UN report of Africa having the most humanitarian needs [6]. Africa suffers disproportionately from climate disasters despite emitting the least greenhouse gases—an irony at the center of a recent international call to action [3]. While complex political factors like aid dependence, western political interests and neo-colonialism have been subjects of debate [28, 29], the geographic and socio-economic vulnerability of the continent are undeniable. According to a climate modeling report, sub-Saharan Africa, especially around the southern belt of the continent, is at high risk for drought, floods, and extreme heat hazards [30]. Furthermore, these hydro-meteorological hazards are anticipated to accelerate due to climate change. The report specified densely populated, poor communities to be at greatest risk, as is present in many countries in sub-Saharan Africa [30]. International humanitarian aid actors must anticipate hydro-meteorological hazards and focus efforts on these specific African communities.

The deadliest earthquake to trigger international aid appeal was the 2003 Bam earthquake in Iran which had a magnitude of 6.5 on the Richter scale. This disaster claimed 26,796 lives,

injured 22,628 others, and damaged 85% of all residential, healthcare, educational and administrative installations. International humanitarian response included 1,600 search-and-rescue and medical teams [31]. Despite this, follow-up evaluation identified inadequate search-and-rescue leading to hypothermia and asphyxiation as a significant factor contributing to mortality [32]. Specifically, local search-and-rescue teams were absent while international search-and-rescue teams arrived too late [33]. According to a UN OCHA report, better local-international coordination through the UN–including specific "trigger mechanisms" for international aid and incorporation of local forces–was required [34]. In a more recent review of the 2011 East Japan earthquake, similar lessons on coordination were echoed, and immigration simplification for international search-and-rescue teams was specifically suggested [35]. International humanitarian aid providers should preemptively establish expedited entry of search-and-rescue teams with local border authorities, particularly with at-risk countries in Asia like Iran, Afghanistan, Indonesia, Turkey, and Tajikistan, which account for more than half of international aid appeals for earthquake response.

Volcanic eruptions were highly associated with international humanitarian aid appeals. One of the largest eruptions in the last 30 years was the January 2022 underwater Hunga Tonga-Hunga Ha'apai volcano eruption near the island of Tonga. This disaster measured an explosive equivalent of nearly 10 megatons of TNT and registered damage equivalent to 18.5% of Tonga's gross domestic product [36, 37]. Over 100,000 people were affected by the resulting tsunamis, disrupted telecommunications lines, contaminated drinking water, dead livestock, and ash-fall several centimeters thick over hundreds of kilometers [38]. The government of Tonga requested aid through the UN Resident Coordinator and was met by an international response that included US $2.7 million and 44 pallets of supplies by UNICEF [36], sanitation programs by IFRC [39], and others. Volcano disaster management leans heavily on mitigation and preparation–strengthening early warning systems, telecommunication infrastructure, community education–but response involves specialized skills and equipment that international humanitarian actors must possess [38]. Delivery of life-saving provisions like safe water and food are disrupted by dangerous flight conditions through airborne ash and on ash-covered runways [40–42]. Health hazards are similarly unique: ash particles and volcanic emissions causing ocular foreign bodies, acute respiratory morbidity, and chronic respiratory complications [43]. Alarmingly, further details are lacking to address volcanic pollutants and their health effects [44]. Nevertheless, international humanitarian aid providers should capitalize on collective experience and existing recommendations to emphasize immediate sea or land access, deploy appropriate gas and particulate masks, and tailor health intervention to victims with pre-existing respiratory diseases [44, 45].

Locust infestations, though infrequent, are disasters likely to involve international aid appeal. Perhaps the most historically significant infestation was the 2004 West African locust plague wherein countries in the subcontinent lost 50 to 90 percent of pasture land, and suffered an estimated US $2.5 billion in crop damage [46]. The UN Food and Agricultural Organization takes the lead in preventing locust infestations [47] by predicting and tracking locust activity ecologically, issuing early warning systems, and advising targeted pesticide dispersal during dormant and juvenile stages [47, 48]. Despite these opportunities, strained local governments have historically failed to prevent locust infestations [49]. It is then probable that other international humanitarian organizations may become instrumental to provide food aid and fund locust control [46]. The "lethargic" and "muted" response to the 2004 West African locust plague highlighted the harm of misinformation (the government of an affected community was criticized for denying the famine) and misdirected media [50, 51]. International humanitarian organizations should source and share accurate, independent, and objective information for a needs-driven response.

Some limitations are inherent in this study. Our analysis relies on the definition employed by the EM-DAT authors. In particular, defining international aid appeal as "any request for international assistance" [16] may leave room for interpretation and possibly overestimate damage reports or underestimate under-represented disasters. The unspecified source of appeal may inadvertently include non-government requests alongside government appeals and hence over-represent comparatively minor disasters. Additionally, the inclusion criteria used by the database authors to define a disaster varies with other databases resulting in limited generalizability [52]. Another limitation inherent to databases is possible data-entry errors introducing inaccuracy to our analysis. These risks are mitigated by the many credible sources informing the EM-DAT and our corroborative gray literature search of news and financial reports from large international humanitarian organizations.

Data from the EM-DAT does not fully capture the characteristics of each disaster and how it affects a population. For example, socioeconomic trends evolving in a community can alter vulnerability to disasters and are difficult to represent solely by geographical data [53]. Similarly, mental health impact in a disaster cannot be accurately quantified by population-level and economic data [54]. Context-specific systematic research on disasters triggering international aid appeal will further guide preparation for international humanitarian aid.

## Conclusions

This study identified disaster and population factors associated with international humanitarian aid appeal. These factors can enable providers to tailor preparation for a needs-driven response upon request by disaster-stricken communities. Depending on the mandate of each organization, preparation should cover climate-vulnerable regions and hydro-meteorological hazards in Africa, include specific equipment, skills, and strategy for earthquakes, volcanic eruptions, and locust infestations, and anticipate large numbers of affected people and fatalities.

## Supporting information

**S1 File.**
(CSV)

## Acknowledgments

Centre for Research on the Epidemiology of Disasters for maintaining the Emergency Events Database (EM-DAT) and providing advice.

## Author Contributions

**Conceptualization:** Lenard Cheng, Attila J. Hertelendy, Alexander Hart, Ryan Hata, Georgina Nouaime, Lina Echeverri, Amalia Voskanyan.

**Data curation:** Lenard Cheng, Lawrence Siu-Chun Law.

**Formal analysis:** Lenard Cheng, Lawrence Siu-Chun Law.

**Investigation:** Lenard Cheng, Lawrence Siu-Chun Law.

**Methodology:** Lenard Cheng, Attila J. Hertelendy, Alexander Hart, Lawrence Siu-Chun Law.

**Project administration:** Lenard Cheng.

**Resources:** Lenard Cheng.

**Supervision:** Attila J. Hertelendy.

**Writing – original draft:** Lenard Cheng, Lawrence Siu-Chun Law, Ryan Hata, Georgina Nouaime.

**Writing – review & editing:** Lenard Cheng, Attila J. Hertelendy, Alexander Hart, Fadi Issa, Lina Echeverri, Amalia Voskanyan, Gregory R. Ciottone.

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
