## [Decision Letter · Decision Letter 0]

4 Apr 2023

PONE-D-23-05682Factors associated with international humanitarian aid appeal for disasters from 1995 to 2015: a retrospective database studyPLOS ONE

Dear Dr. Cheng,

Thank you for submitting your manuscript to PLOS ONE. After careful consideration, we feel that it has merit but does not fully meet PLOS ONE’s publication criteria as it currently stands. Therefore, we invite you to submit a revised version of the manuscript that addresses the points raised during the review process.

We look forward to receiving your revised manuscript.

Kind regards,

Dr. Md Nazirul Islam Sarker

Academic Editor

PLOS ONE

Journal Requirements:

Additional Editor Comments (if provided):

The author is advised to address all comments point-by-point.

Reviewers' comments:

Reviewer's Responses to Questions

**Comments to the Author**

1. Is the manuscript technically sound, and do the data support the conclusions?

Reviewer #1: Yes

Reviewer #2: Yes

2. Has the statistical analysis been performed appropriately and rigorously? 

Reviewer #1: Yes

Reviewer #2: Yes

3. Have the authors made all data underlying the findings in their manuscript fully available?

Reviewer #1: Yes

Reviewer #2: Yes

4. Is the manuscript presented in an intelligible fashion and written in standard English?

Reviewer #1: Yes

Reviewer #2: Yes

5. Review Comments to the Author

Reviewer #1: The aim of this study is to identify dusaste and population factors associated with international aid appeal.　Results were that aid appeals were more likely to be triggered by disasters which killed more people, affected more people, and occurred in Africa and earthquakes, volcanic activity, and insect infestations. The authors concluded that humanitarian aid providers can focus preparation on these identified factors.

The study and results are interesting. The reviewer could not find major concerns on the aim of this study, study design, study methods, analysis, results, and the conclusion based on the results.

The reviewer recommends being published after minor revision.

・Page 6, line 139- 141; legend of figure is inserted into the body of manuscript.

・Page 14, line 275-277; the number (1)~(3) are confusing with the number for references.

Reviewer #2: This is a well written manuscript. The introduction should address more on the unintended consequences of international aid. Doesn't it do more harm than good. Does it have any impact of the affected communities. Is it not another way of western World continued manipulation of the so called poor (even when we know they are not poor), but called poor to attract funding. Shouldn't we let communities help themselves instead of coming up as saviors.

Discuss these points further:

Dependence: One of the most significant disadvantages of international aid is that it can create a culture of dependence in recipient countries. If countries become reliant on foreign aid, they may not develop their own economies, which can hinder long-term growth and development.

Corruption: International aid can also lead to corruption, as governments or organizations may use the funds for their own purposes rather than the intended recipients. This can prevent aid from reaching those who need it most and may even exacerbate the problems it was intended to solve.

Disruption of Local Markets: International aid can disrupt local markets by flooding them with cheap or free goods, which can harm local businesses and prevent economic growth.

Conditionality: Many countries that provide international aid may attach conditions to it, such as requiring the recipient country to adopt specific economic policies or make political changes. This can limit a country's sovereignty and prevent it from making its own decisions.

Unsustainable Solutions: Aid programs may be designed to address immediate needs, such as providing food and medicine during a crisis. However, they may not be sustainable in the long term and may not address the root causes of poverty or underdevelopment.

My own thought is that aid should be stopped. I wish this paper had made this recommendation. It is disempowering and manipulative whether in disaster or not. It is actually immoral for one country to claim to help another. There is no disabled country on earth, give them a chance to evolve and grow.

Overall, the methods section appears to be well-written and provides sufficient detail on the data source, inclusion/exclusion criteria, and statistical analysis. However, there are a few areas that could be improved or clarified:

Inclusion/exclusion criteria: While the inclusion/exclusion criteria are clearly stated, it may be helpful to provide some justification for why these specific criteria were chosen. For example, why was the threshold for human deaths set at 10? Why was the year 1995 chosen as the starting point for the search? and why stopped 2015.

Missing data: The authors acknowledge that missing data was present in the dataset, but it is not clear how they handled this issue. While the use of multiple imputation is mentioned, it is not clear how this method was implemented or what software was used.

Logistic regression: The authors state that logistic regression was used to examine the independent association between factors and international aid appeal, but it is not clear what specific factors were included in the regression model. Providing a list of these factors would help readers understand the analysis more clearly.

Ethics approval: While the authors state that ethics approval was not required, it may be helpful to provide some additional information on the ethical considerations related to the use of publicly available data, particularly given the sensitive nature of the topic (i.e. disasters and deaths).

The Results section of this manuscript presents findings on 13,961 disasters recorded from 1995 to 2015, of which 168 (1.2%) involved international aid appeals. The authors use descriptive statistics and inferential analyses to explore the association between disaster characteristics and international aid appeals. The section is well-organized, clear, and easy to understand, and it presents the findings in a logical and coherent way.

Overall, the discussion is well-researched and provides a detailed analysis of the association between disasters and international aid appeals. The authors provide several examples to support their argument and highlight the importance of sustainability, coordination, and anticipation of hydro-meteorological hazards.

However, I am surprised by the findings about Africa. I believe that most of the appeals for aid from Africa are not made by Africans themselves. In reality, Africa has been one of the least affected continents in recent times, as evidenced by the limited impact of the COVID-19 pandemic. Africa reported very few major disasters in recent years. I think this misconception is rooted in the DNA of donor organizations, who continue to mobilize resources from Africa and portray it as the most needy continent, even though it is not. This data needs to be thoroughly interrogated.

6. PLOS authors have the option to publish the peer review history of their article (what does this mean?). If published, this will include your full peer review and any attached files.

Reviewer #1: No

Reviewer #2: **Yes: **Luke Nyakarahuka

While revising your submission, please upload your figure files to the Preflight Analysis and Conversion Engine (PACE) digital diagnostic tool, https://pacev2.apexcovantage.com/. PACE helps ensure that figures meet PLOS requirements. To use PACE, you must first register as a user. Registration is free. Then, login and navigate to the UPLOAD tab, where you will find detailed instructions on how to use the tool. If you encounter any issues or have any questions when using PACE, please email PLOS at figures@plos.org. Please note that Supporting Information files do not need this step.<quillbot-extension-portal></quillbot-extension-portal>

---

## [Author Response · Author response to Decision Letter 0]

13 Apr 2023

Reviewer #1:

 1. Page 6, line 139- 141; legend of figure is inserted into the body of manuscript.

Thank you for pointing this out. Following PLOS ONE guidelines, (https://journals.plos.org/plosone/s/figures), we have removed Figure label, title, and legends from the TIFF file, and kept these in the manuscript text in read order. 

 2. Page 14, line 275-277; the number (1)~(3) are confusing with the number for references

Thank you for pointing this out. We have removed the numbered points in the prose to avoid confusion. 

Reviewer #2:

 3. Discuss these points further:

 Dependence: One of the most significant disadvantages of international aid is that it can create a culture of dependence in recipient countries. If countries become reliant on foreign aid, they may not develop their own economies, which can hinder long-term growth and development.

 Corruption: International aid can also lead to corruption, as governments or organizations may use the funds for their own purposes rather than the intended recipients. This can prevent aid from reaching those who need it most and may even exacerbate the problems it was intended to solve.

 Disruption of Local Markets: International aid can disrupt local markets by flooding them with cheap or free goods, which can harm local businesses and prevent economic growth.

 Conditionality: Many countries that provide international aid may attach conditions to it, such as requiring the recipient country to adopt specific economic policies or make political changes. This can limit a country's sovereignty and prevent it from making its own decisions.

 Unsustainable Solutions: Aid programs may be designed to address immediate needs, such as providing food and medicine during a crisis. However, they may not be sustainable in the long term and may not address the root causes of poverty or underdevelopment.

 My own thought is that aid should be stopped. I wish this paper had made this recommendation. It is disempowering and manipulative whether in disaster or not. It is actually immoral for one country to claim to help another. There is no disabled country on earth, give them a chance to evolve and grow.

Thank you for these valued opinions. We share the Reviewer’s concern against uninvited, unregulated, and intrusive international humanitarian action. This shared mission is in fact the motivation for this project. As a scientific manuscript, we base our discussion on available data and attempt to best balance our opinions with the understandably varied views and experiences of the international community. To bolster our discussion with the Reviewer’s valued opinions, we added a forerunning paragraph to the effect of the Reviewer’s comments by referencing PAHO and WHO publications. 

 4. Inclusion/exclusion criteria: While the inclusion/exclusion criteria are clearly stated, it may be helpful to provide some justification for why these specific criteria were chosen. For example, why was the threshold for human deaths set at 10? Why was the year 1995 chosen as the starting point for the search? and why stopped 2015.

Thank you for evaluating this point. A literature search and subject expertise does not yield specific justification for the database author’s inclusion criteria, although it is widely accepted. We added discussion on this as a limitation and referenced a recent article which arrived at the same conclusion. We added elaboration in the methods section on the rationale behind limiting the search until 2015. The rationale for 1995 as a start point is also elaborated in the methods section.

 5. Missing data: The authors acknowledge that missing data was present in the dataset, but it is not clear how they handled this issue. While the use of multiple imputation is mentioned, it is not clear how this method was implemented or what software was used.

Thank you for requesting clarification. We added the specifications for the multiple imputation into the methods section in the main text: (Fully conditional specification, 10 iterations, predictive mean matching, 10 imputations, SPSS 26.0).

 6. Logistic regression: The authors state that logistic regression was used to examine the independent association between factors and international aid appeal, but it is not clear what specific factors were included in the regression model. Providing a list of these factors would help readers understand the analysis more clearly.

Thank you for requesting clarification. We added underlines to all variables in Table 1 which were entered into logistic regression. We also added a sentence in the legend of Table 1: Significant p-values were underlined and the corresponding variables were entered in the logistic regression.

 7. Ethics approval: While the authors state that ethics approval was not required, it may be helpful to provide some additional information on the ethical considerations related to the use of publicly available data, particularly given the sensitive nature of the topic (i.e. disasters and deaths).

Thank you for requesting additional information. We elaborated in the methods section on the primary reasons that ethics approval was not required, and made special effort to address potentially sensitive factors like deaths and injury. Our main justification is the non-individual patient level data, and the publicly available data source.

---

## [Decision Letter · Decision Letter 1]

17 May 2023

Factors associated with international humanitarian aid appeal for disasters from 1995 to 2015: a retrospective database study

PONE-D-23-05682R1

Dear Dr. Cheng,

We’re pleased to inform you that your manuscript has been judged scientifically suitable for publication and will be formally accepted for publication once it meets all outstanding technical requirements.

Kind regards,

Md Nazirul Islam Sarker

Academic Editor

PLOS ONE

Additional Editor Comments (optional):

The author is requested to keep in touch with the production team for the remaining publication process.

Reviewers' comments:

Reviewer's Responses to Questions

**Comments to the Author**

1. If the authors have adequately addressed your comments raised in a previous round of review and you feel that this manuscript is now acceptable for publication, you may indicate that here to bypass the “Comments to the Author” section, enter your conflict of interest statement in the “Confidential to Editor” section, and submit your "Accept" recommendation.

Reviewer #1: All comments have been addressed

2. Is the manuscript technically sound, and do the data support the conclusions?

Reviewer #1: Yes

3. Has the statistical analysis been performed appropriately and rigorously? 

Reviewer #1: Yes

4. Have the authors made all data underlying the findings in their manuscript fully available?

Reviewer #1: Yes

5. Is the manuscript presented in an intelligible fashion and written in standard English?

Reviewer #1: Yes

6. Review Comments to the Author

Reviewer #1: After revision, there is no revision pointed out by the reviewer. The reviewer recommends publication.

7. PLOS authors have the option to publish the peer review history of their article (what does this mean?). If published, this will include your full peer review and any attached files.

Reviewer #1: No

---

## [Editor Report · Acceptance letter]

22 May 2023

PONE-D-23-05682R1 

Factors associated with international humanitarian aid appeal for disasters from 1995 to 2015: a retrospective database study 

Dear Dr. Cheng:

I'm pleased to inform you that your manuscript has been deemed suitable for publication in PLOS ONE. Congratulations! Your manuscript is now with our production department. 

Kind regards, 

on behalf of

Dr. Md Nazirul Islam Sarker 

Academic Editor

PLOS ONE